# The gut microbiome but not the resistome is associated with urogenital schistosomiasis in preschool-aged children

Derick N.M. Osakunor [1 ✉], Patrick Munk [2], Takafira Mduluza[3], Thomas N. Petersen[2], Christian Brinch[2], Alasdair Ivens[1], Theresa Chimponda[3], Seth A. Amanfo[4,5], Janice Murray[1,5], Mark E.J. Woolhouse [4,5], Frank M. Aarestrup [2] & Francisca Mutapi[1,5]

Helminth parasites have been shown to have systemic effects in the host. Using shotgun metagenomic sequencing, we characterise the gut microbiome and resistome of 113 Zimbabwean preschool-aged children (1–5 years). We test the hypothesis that infection with the human helminth parasite, *Schistosoma haematobium*, is associated with changes in gut microbial and antimicrobial resistance gene abundance/diversity. Here, we show that bacteria phyla *Bacteroidetes, Firmicutes, Proteobacteria*, and fungi phyla *Ascomycota, Microsporidia, Zoopagomycota* dominate the microbiome. The abundance of *Proteobacteria, Ascomycota*, and *Basidiomycota* differ between schistosome-infected versus uninfected children. Specifically, infection is associated with increases in *Pseudomonas, Stenotrophomonas, Derxia, Thalassospira, Aspergillus, Tricholoma*, and *Periglandula*, with a decrease in *Azospirillum*. We find 262 AMR genes, from 12 functional drug classes, but no association with individual-specific data. To our knowledge, we describe a novel metagenomic dataset of Zimbabwean preschool-aged children, indicating an association between urogenital schistosome infection and changes in the gut microbiome.

[1] Institute of Immunology & Infection Research, University of Edinburgh, Ashworth Laboratories, King's Buildings, Charlotte Auerbach Road, Edinburgh EH9 3FL, UK. [2] Research Group for Genomic Epidemiology, National Food Institute, Technical University of Denmark, DK-2800 Kongens Lyngby, Denmark. [3] Biochemistry Department, University of Zimbabwe, P.O. Box MP167 Mount Pleasant, Harare, Zimbabwe. [4] Usher Institute of Population Health Sciences & Informatics, University of Edinburgh, Ashworth Laboratories, Kings Buildings, Charlotte Auerbach Road, Edinburgh EH9 3FL, UK. [5] NIHR Global Health Research Unit Tackling Infections to Benefit Africa (TIBA), University of Edinburgh, Ashworth Laboratories, King's Buildings, Charlotte Auerbach Road, Edinburgh EH9 3FL, UK. ✉email: d.osakunor@ed.ac.uk

The human gut comprises a diverse ecosystem of microbes, predominantly bacteria, in addition to viruses, fungi and other eukaryotes[1]. Evidence shows that humans rely on the symbiotic relationship with the resident microbial taxa present in humans (microbiota) for extracting essential nutrients from food, as a first line of protection from pathogens, and as a mechanism for shaping the immune system[2]. Shotgun metagenomic sequencing has allowed characterisation of the microbiome (the assembly of genomes of the microbiota) among different human populations, showing considerable heterogeneity[2,3]. Populations in Africa have been underrepresented in such studies, with a major focus on Western populations[4]. Other studies have included diverse but older populations, not allowing the factors inherent to African childhood to be fully disentangled[5–8]. Given the potential window of opportunity for influencing health through the microbiome in infants and young children[9], research focus on this age group is important. Findings from consortiums including the Human Heredity and Health in Africa (H3 Africa) and the HapMap Project will be invaluable for informing nutraceuticals in Africa[10,11].

The composition of the gut microbiome is influenced by age[8,12], diet and geography[5,13,14], host genotype[15], exposure to maternal microbiota[16], as well as environmental factors[17] including the role of protozoal and helminth parasites[18]. In Africa, children are exposed to several acute and chronic parasitic infections that can impact children's growth and development[19]. In particular, helminth parasites (as shown for schistosome worms) can be contracted by children as young as 6-month-old or less[20,21], and these can persist into the second decade of life where they modulate the immune system as well as cause morbidity and pathology[22]. In very young children, the gut microbial population continues to evolve until about age 3–5 years[8,12], thus it is important to establish how external factors, especially infections, that young children are exposed to, influence the microbiome.

Schistosomiasis is a disease caused by infection with trematodes of the genus Schistosoma—the predominant human species found in Africa being the urogenital (S. haematobium) and intestinal (S. mansoni) forms[23]. Pathology from the disease is mostly from immunological reactions to trapped eggs attempting to migrate through to the bladder or intestinal lumen, depending on the species involved. The infection causes immunomodulatory effects which help to promote both parasite and host survival[24,25], and in preschool-aged children, consequences can extend to malnutrition, poor growth and cognition, reduced vaccine efficacy, and altered prognosis of co-infections[26,27]. Treatment for schistosomiasis is through administration of the antihelminthic drug, praziquantel, which is effective against all schistosome species[28].

A number of experimental and human studies, including our own, have examined the association between helminth infections and the structure and composition of the gut microbiome[29–35]. It has been suggested that the immunomodulatory effects of schistosome infection can extend to the gut microbiota through direct intestinal or systemic interactions[18]. Work in experimental models shows that depletion of the gut bacteria is associated with reduced S. mansoni egg excretion, gut pathology and inflammation[32]. Recently, fluctuations in the composition of the gut microbiota of mice infected with S. mansoni, before and after intestinal damage from egg transmission was shown[36]. This is consistent with a role of the mammalian gut microbiota in the pathogenesis of schistosome infection. However, unlike S. mansoni (intestinal form) that inhabits the same environment as the gut microbiota, S. haematobium predominantly resides in the venous plexus of the bladder (although occasionally in the mesenteric circulation[37]), and thus presents a need to study the indirect systemic impacts of infection on the gut microbiota—a more likely interaction. Correlations between the gut microbiome and systemic diseases such as rheumatoid arthritis suggest the importance of such systemic interactions[38]. Phenotypic and mechanistic studies on systemic interactions between helminths and the microbiome in natural human infections are still in their infancy, and more studies are needed. In a previous study, we found differences in the gut microbiome between S. haematobium infected versus uninfected children, aged up to 13 years old[35]. This was supported by Schneeberger et al.[34], suggesting that genetic and environmental factors may play an additional role. Recently, a study conducted among older children (11–15 years) in Nigeria showed that urogenital schistosomiasis is associated with disruptions in the gut microbiome, suggesting that this may be a further consequence of schistosome infection[39]. However, substantial knowledge gaps on the interaction between the gut microbiome and Schistosoma infection in preschool-aged children still exist. The biggest challenges are demonstrating causation and elucidating mechanistic pathways for any existing interactions.

In addition to the schistosome–microbiota interactions, other interactions relevant to the health of the host have also been reported to occur within the gut ecosystem. One such example is the ability of Salmonella to persist in the body by attaching to intestinal schistosomes, evading repeated antibiotic treatments, increasing the Salmonella population and eventually, potential antibiotic resistance[40,41]. Furthermore, the microbiome is a reservoir for antimicrobial resistance (AMR) genes (resistome)[42,43], and provides an ideal environment for AMR gene exchange among the "resident" and transitory bacterial population[44]. Such interactions are likely to impact the structure and diversity of the resident microbial population, as well as the overall AMR gene composition. We therefore investigated the structure of AMR genes to bacteria and determined if this was associated with any host-related factors including socio-demography, antibiotic use, current schistosome infection, as well as feeding, growth and nutritional indices. AMR remains one of the largest threats to human health, with numerous calls for antibiotic resistance stewardship worldwide[42,45,46]. However, this population of African preschool-aged children is understudied, and almost all AMR gene studies are conducted in industrialised settings[43,47–50]. Such settings contrast with low-and middle-income countries in terms of access to safe water and sanitation, and access to antibiotics, with or without prescriptions[51].

Within the framework of a larger paediatric schistosomiasis study in Zimbabwe, the present study focuses on shotgun metagenomic sequencing of stool samples from preschool-aged children, aged 1–5 years old. To add to the repository of information on the gut microbiome and AMR studies in this young population, we characterise the structure and diversity of the gut microbiome (to include the fungi repertoire) and resistome. We apply these data to test the hypothesis that S. haematobium infection is associated with alterations in the gut microbial and AMR gene abundance and diversity. We find that the microbiome but not the resistome is associated with S. haematobium infection, independent of age, sex and village.

## Results

**Population characteristics**. Of the 113 participants included in the study, the mean age was $3.7 \pm 1.1$ years, of which 56 were females (49.6%). Sixty-eight (60.2%) and 45 (39.8%) children were from Chihuri and Mupfure villages, respectively. Antimicrobial use data showed that 58 (51.3%) participants had received antibiotics [amoxicillin (31), co-trimoxazole (27), both (9)], while 18 (15.9%) had not; no information was obtained for the remaining 37 (32.7%) participants. Previous history of praziquantel treatment (for schistosome infection) was reported

among 29/105 (27.6%) children. *S. haematobium* infection prevalence was 15.9% (18/113), with mean infection intensity of 1.79 eggs/10 ml urine (SEM = 0.76; range = 0–74).

We gathered data on the history of feeding habits and nutritional status of the children. The majority (83.6%) were breastfed, with duration ranging from 2 to 48 months (median = 18 months IQR: 17–20). Children were introduced to solid foods between 1 and 24 months after birth. Diet comprised mainly of traditional maize flour porridges (97%; 96/103), the commercial Cerelac® porridge (1.9%; 2/103), and potatoes (1%; 1/103). Anthropometric measures, adjusted for age, were used to assess nutritional status[27]. Based on the weight-for-height $Z$ scores (WHZ), 3.7% (4/107) of individuals were malnourished and 14.7% (16/109) were stunted, based on the height-for-age $Z$ scores (HAZ)[52].

**Taxonomic composition of the microbiome**. The number of classified read pairs per sample ranged from 3,994,704 to 13,164,482. An average 45.1% of read pairs were mapped to specific reference sequences in the genomic database; this is similar to other studies with the proportion of unmapped reads ranging from 42% to 68%[53–55]. At any taxonomic level, a putative taxonomic classification could not be assigned to at least 33% of the mapped read pairs and were thus classified as "Unknown".

In the 113 stool samples, 845 bacteria genera (from 20 unique phyla) and 228 fungi genera (from six unique phyla) were detected. As shown in Fig. 1, the most abundant bacteria phyla in decreasing order were *Bacteroidetes* (genera: *Prevotella, Bacteroides, Alistipes*), *Firmicutes* (genera: *Eubacterium, Faecalibacterium, Clostridium,*

*Roseburia*), and *Proteobacteria* (genus: *Succinatimonas*). The most abundant fungi phyla were *Ascomycota* (genera: *Protomyces, Aspergillus, Taphrina, Saccharomyces, Candida, Nakaseomyces*), *Microsporidia* (genus: *Enterocytozoon*), and *Zoopagomycota* (genus: *Entomophthora*) [Fig. 2]. These phyla dominated the microbiome and were present in all samples.

**Variation in the microbiome and association with sample metadata**. Principal component analysis (PCA) was used to initially examine variability and patterns in the data set across the first two principal components. At the phylum level, PCA explained 62% and 42.0% of the total variation in fungi and bacteria, respectively. At the genus level, however, PCA explained 34% and 16% of the total variation in fungi and bacteria respectively. The model showed homogeneity in components with no distinct clustering according to metadata and may reflect a high diversity in the cohort. PCA plots and cluster dendrograms for bacteria and fungi content per sample is shown in Supplementary Figs. 1–3.

Permutational multivariate analysis of variance (PERMANOVA) analysis showed a significant effect of age (false discovery rate (FDR) = 0.024) and village (FDR = 0.039) [details shown in Supplementary Fig. 4], schistosome infection status (FDR = 0.039) and schistosome infection intensity (FDR = 0.012) on bacteria genera, across samples. There was also a significant effect of schistosome infection status (FDR = 0.006) and schistosome infection intensity (FDR = 0.006) on fungi genera, across the samples. For both bacteria and fungi genera, no such effects were found for sex, nutritional and growth variables, feeding, previous

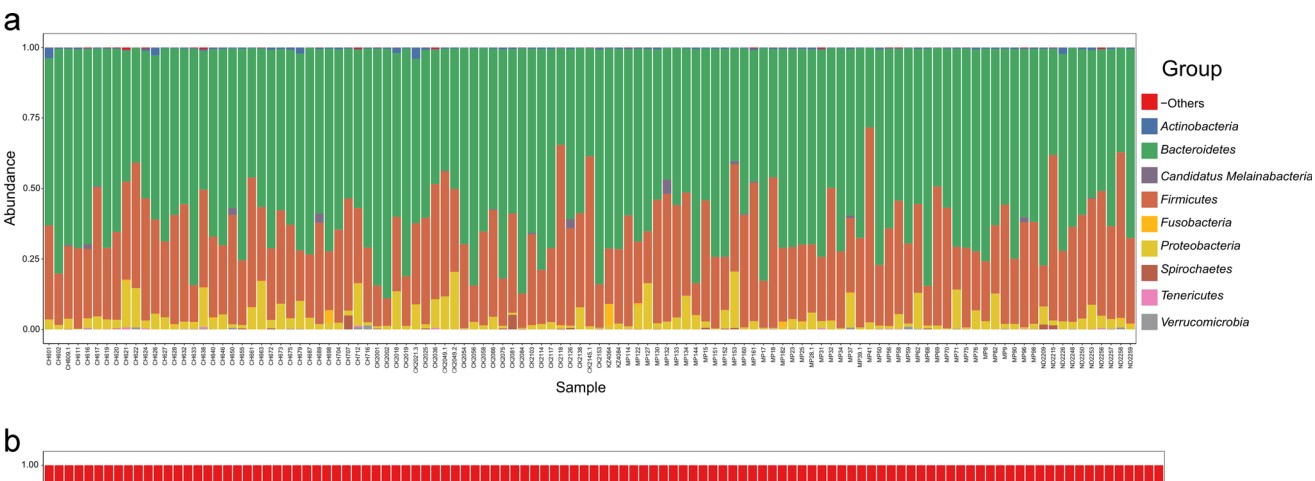

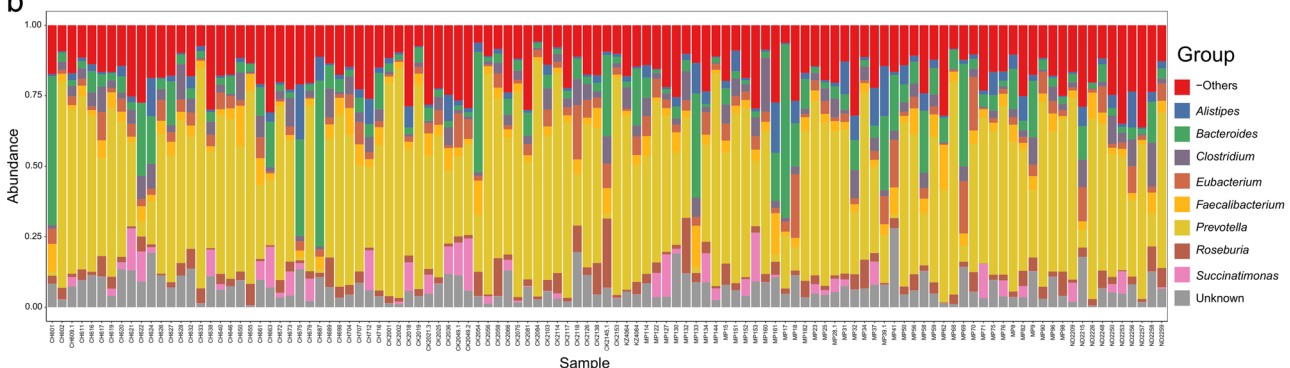

**Fig. 1 Overview of bacterial microbiota abundance and composition.** From read mapping to the genomic database, abundance was calculated for each microbial taxa across all samples. Stacked bar charts show the most abundant bacteria (**a**) phyla and (**b**) genera per sample, proportional to the total microbiota within each sample (*n* = 113 biologically independent samples). Charts were generated using normalised, zero-corrected abundance matrices. "Unknown" represents abundance data for which a putative taxonomic classification could not be assigned. "-Others" represents abundance data for all other taxa in the abundance data set.

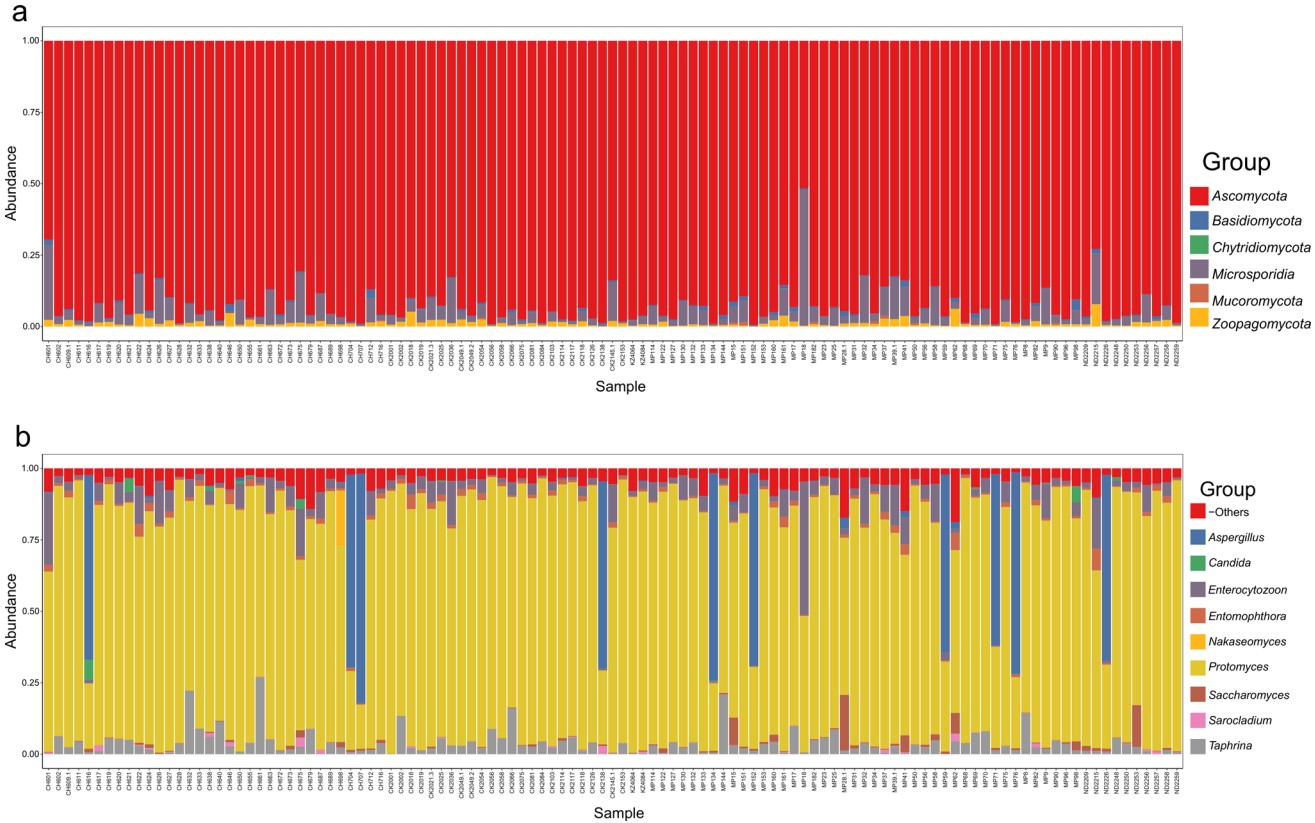

**Fig. 2 Overview of fungal microbiota abundance and composition.** From read mapping to the genomic database, abundance was calculated for each microbial taxa across all samples. Stacked bar charts show the most abundant fungi (**a**) phyla and (**b**) genera per sample, proportional to the total microbiota within each sample (*n* = 113 biologically independent samples). Charts were generated using normalised, zero-corrected abundance matrices. "-Others" represents abundance data for all other taxa in the abundance data set.

praziquantel treatment, and antibiotic use (FDR > 0.05). Summary output from the analysis is shown in Table 1.

**Different genera by schistosome infection status and intensity.** We investigated further, via analysis of composition of microbiomes (ANCOM), how specific bacteria and fungi genera were associated with *S. haematobium* infection, while controlling for age, sex and village, followed by evaluation for association with infection intensity. In total, eight genera were identified, five from bacteria (*Pseudomonas*: $W = 347$, *Azospirillum*: $W = 346$, *Stenotrophomonas*: $W = 292$, *Derxia*: $W = 288$, and *Thalassospira*: $W = 292$) and three from fungi (*Aspergillus*: $W = 75$, *Tricholoma*: $W = 73$ and *Periglandula*: $W = 70$). The magnitude of these changes were shown by plotting the abundance of each sample to highlight differences between groups. In schistosome-positive children, the abundance of all but *Azospirillum* was higher (Fig. 3a–e). This observation was consistent with infection intensity [*Pseudomonas* ($r = 0.3$; $p = 0.001$), *Stenotrophomonas* ($r = 0.4$; $p < 0.001$), *Derxia* ($r = 0.6$; $p < 0.001$), *Thalassospira* ($r = 0.6$; $p < 0.001$) and *Azospirillum* ($r = -0.4$; $p < 0.001$)] as shown in Fig. 3f–j. Likewise, the abundance of *Aspergillus*, *Tricholoma*, and *Periglandula* was higher in schistosome-positive children (Fig. 4a–c) and was consistent with infection intensity as shown in Fig. 4d–f [*Aspergillus* ($r = 0.5$; $p < 0.001$), *Tricholoma* ($r = 0.5$; $p < 0.001$), and *Periglandula* ($r = 0.4$; $p < 0.001$)].

**AMR gene characterisation.** An average 0.06% of read pairs were mapped to AMR genes in the ResFinder database. We found evidence of 262 AMR genes, belonging to 12 functional drug class levels. AMR genes belonging to tetracycline was the most

common, followed by beta-lactam, macrolide, sulfonamide and nitroimidazole. Of these, the most abundant genes were *cfxA6*, followed by *tet(Q)*, *tet(W)*, *sul2*, *erm(F)* and *nimE* (Fig. 5).

**Variation in the resistome and association with sample metadata.** PCA was used to initially examine variability and to identify clustering according to individual metadata. The model for the first two components explained 18.0% and 48.0% of the total variability in AMR genes and drug classes respectively. Similarly, there was no clustering according to individual metadata, reflecting high cohort diversity and the role of other factors in influencing the resistome. PCA plots and cluster dendrograms of AMR genes and their drug classes per sample is shown in Supplementary Figs. 5–6.

PERMANOVA analysis did not show any significant association of AMR genes with age, village, sex, feeding, malnutrition, stunting, *S. haematobium* infection, previous praziquantel treatment and antibiotic use on AMR genes. Model summaries of sample metadata and association with AMR genes is shown in Supplementary Data 2.

**Discussion**
Using shotgun metagenomic sequencing, we characterised the structure and composition of the human gut microbiome and resistome in this Zimbabwean preschool population (≤5 years old). Age[8,12], dietary and environmental patterns[5,13,17], ethnicity, and geography[14,56] have a substantial impact on the taxonomic composition of the microbiome. *Prevotella* and *Candida*[57,58] have been associated with carbohydrate-rich diets, and *Bacteroides* with protein-rich diets[57]. This is a reflection of the dietary lifestyle

**Table 1 Model summaries of sample metadata and association with the gut microbiome.**

| Variable | n | Bacteria | | | | Fungi | | | |
|---|---|---|---|---|---|---|---|---|---|
| | | p value | SSExplain | SSTotal | FDR | p value | SSExplain | SSTotal | FDR |
| Age (years) | 113 | 0.004 | 1344.6 | 82733.4 | 0.024 | 0.082 | 128.6 | 9489.2 | 0.197 |
| Sex | 113 | 0.172 | 878.1 | 83200.0 | 0.258 | 0.439 | 82.8 | 9534.9 | 0.671 |
| Village | 113 | 0.012 | 1254.0 | 82824.0 | 0.039 | 0.060 | 140.4 | 9477.4 | 0.180 |
| S.h. infection status (pos/neg) | 113 | 0.013 | 1185.1 | 82892.9 | 0.039 | 0.001 | 339.0 | 9278.7 | 0.006 |
| S.h. infection intensity | 113 | 0.001 | 1514.5 | 82563.6 | 0.012 | 0.001 | 670.1 | 8947.7 | 0.006 |
| Malnourished, yes/no (WHZ) | 107 | 0.866 | 589.1 | 78498.2 | 0.913 | 0.830 | 59.5 | 9145.6 | 0.830 |
| Stunted, yes/no (HAZ) | 109 | 0.407 | 754.6 | 79751.0 | 0.542 | 0.611 | 71.5 | 9227.2 | 0.671 |
| Months breastfed | 90 | 0.082 | 954.9 | 64235.8 | 0.140 | 0.470 | 75.2 | 6985.0 | 0.671 |
| Months solid food introduced | 102 | 0.913 | 573.1 | 75583.0 | 0.913 | 0.615 | 73.5 | 8792.0 | 0.671 |
| Previous praziquantel treatment | 105 | 0.071 | 991.9 | 77387.4 | 0.140 | 0.233 | 101.6 | 8934.5 | 0.466 |
| Amoxicillin (yes/no) | 76 | 0.771 | 646.6 | 55603.4 | 0.913 | 0.531 | 78.4 | 6567.3 | 0.671 |
| Co-trimoxazole (yes/no) | 76 | 0.030 | 1083.6 | 55166.4 | 0.072 | 0.048 | 158.3 | 6487.4 | 0.180 |

The table represents PERMANOVA output for bacteria and fungi genera. Classification of nutritional status was based on a cut off $<-2$ Z scores[52]. Schistosome infection intensity was log transformed ($\log_{10}$ [egg count + 1]). *S. S. haematobium*, *WHA* weight-for height Z scores, *HAZ* height-for-age Z scores, *pos/neg* positive/negative, *p value* unadjusted p value, *FDR* adjusted p value (FDR-corrected), *SSExplain* explained sum of squares, *SSTotal* total sum of squares.

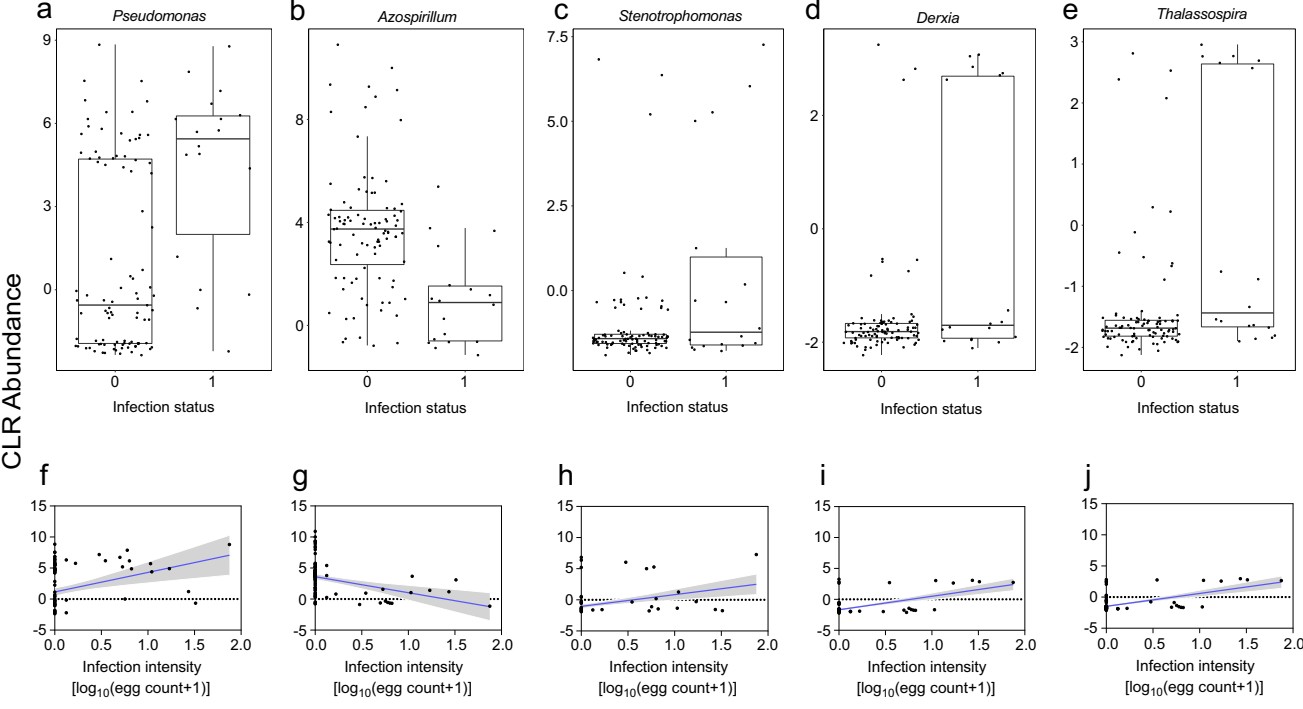

**Fig. 3 Different bacteria genera between schistosome-infected children compared to uninfected children. (a–e)** Box plots showing the abundance of specific bacteria genera, grouped by *S. haematobium* infection status. The horizontal box lines represent the first quartile, the median, and the third quartile. Whiskers denote the range of points within the first quartile −1.5× the interquartile range and the third quartile +1.5× the interquartile range. **(f–j)** Scatter plots showing linear regression analysis of *S. haematobium* infection intensity and bacteria genera abundance. The clr-transformed abundance data were used for all plots. Infection status was coded as 0 and 1 for negative (n = 95) and positive (n = 18), respectively. *S. haematobium* infection intensity was log transformed [$\log_{10}$ (egg count + 1)]. Shaded areas indicate the 95% confidence intervals.

among populations in developing countries[5,17], including infants[59], and these genera were among the most abundant bacteria and fungi genera found in the current study population. Similar to our previous findings[35], we found age but not sex-related associations in bacteria genus diversity. Given that the population in the current study were ≤5 years old, this is consistent with the microbiome being more dynamic in the early years of life before stabilising to a more adult-like state[8,12].

Differences observed in the microbiome between developing and developed countries have been attributed to factors inherent to such developing areas[5], which may include the role of persistent prevalence of helminth infections, as reviewed by

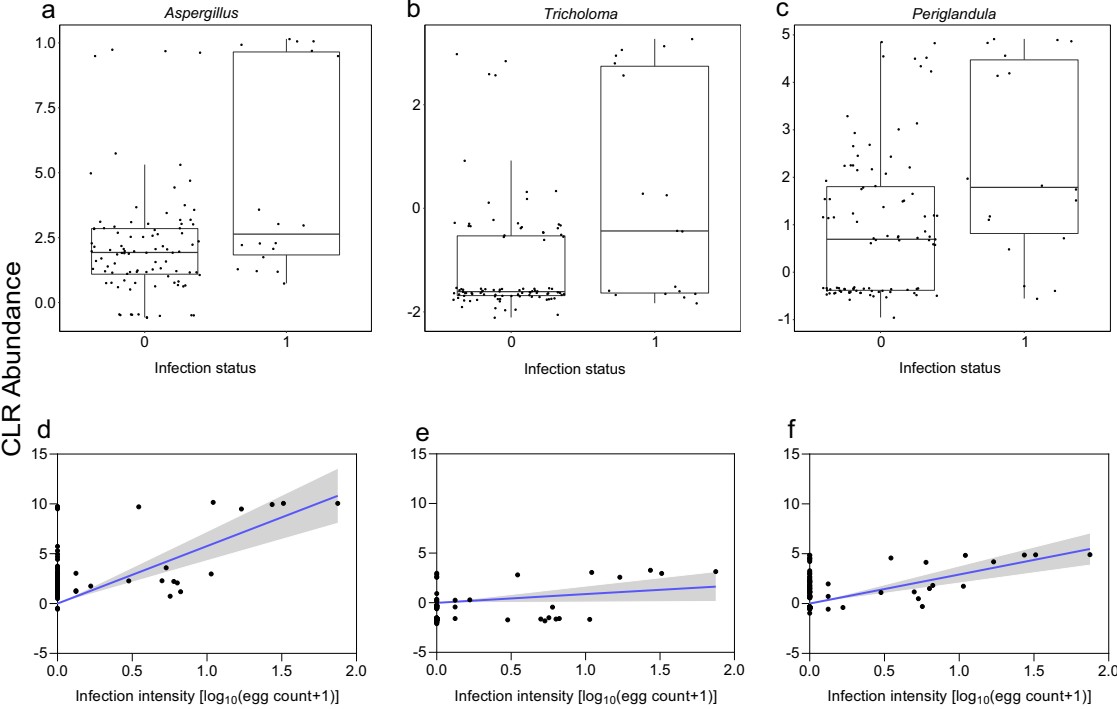

**Fig. 4 Different fungi genera between schistosome-infected children compared to uninfected children. (a–c)** Box plots showing the mean abundance of specific fungi genera, grouped by *S. haematobium* infection status. The horizontal box lines represent the first quartile, the median and the third quartile. Whiskers denote the range of points within the first quartile −1.5× the interquartile range and the third quartile +1.5× the interquartile range. (**d–f**) Scatter plots showing linear regression analysis of *S. haematobium* infection intensity and fungi genera abundance. The clr-transformed abundance data were used for all plots. Infection status was coded as 0 and 1 for negative ($n = 95$) and positive ($n = 18$), respectively. *S. haematobium* infection intensity was log transformed [$\log_{10}$ (egg count + 1)]. Shaded areas indicate the 95% confidence intervals.

Mishra et al.[18]. Our findings are consistent with observations that schistosome infection is associated with alterations in the diversity and abundance of specific taxonomic groups in the microbiome[34,35]. In the aforementioned studies, which included preschool and school-aged children, 16S rRNA sequencing showed that *Prevotella* and *Proteobacteria* were more abundant in children infected with *S. haematobium*[35] and *S. mansoni*[34] respectively, when compared to uninfected children. To the best of our knowledge, the novelty of the current study is the fact that this population is much younger (≤5 years old), an age group whose gut microbiome structure is most likely still being established. Our study thus provides an important insight into helminth infection and its association with changes during the establishment of the gut microbiome in preschool-aged children in endemic areas. We also expand on this to include the fungal component of the microbiome. More relevant to our finding is that this association was independent of age, sex and village.

Phyla that clearly differentiated the microbiome of the schistosome-infected versus uninfected children were *Proteobacteria*, *Ascomycota* and *Basidiomycota*. These were among the top five most abundant bacteria or fungi phyla and were present in all samples, thus make a major contribution to the overall microbiome composition. *Proteobacteria* has been shown to be present in lower abundances in healthy individuals, and any increases in abundance of members of this phylum confirm dysbiosis and a link with increased disease risk, progression and burden[60]. Attempts have been made to expand the body of knowledge on the fungi repertoire and diversity in the human microbiome[61] and their association with infection and disease[62–64]. Studies have suggested that gut fungal populations directly or indirectly help to maintain healthy intestinal homoeostasis and that dysbiosis has immunological consequences[65]. Increases in specific

fungi populations such as *Aspergillus* have been associated with increased eosinophil levels[66] and an exaggerated Th2 response[65], also characteristic of schistosome infection, which may explain our observation of the association of schistosome infection with specific fungal populations (*Aspergillus*, *Tricholoma* and *Periglandula*). However, whether our observation was due to primary changes in the fungal population or were secondary to changes in the bacterial population is unclear. To the best of our knowledge, this is the first study to examine such an association and further studies into the role of fungal dysbiosis in schistosome infection are warranted.

Although we cannot infer causation, we are able to determine that for the genera differentiating the microbiome of the schistosome-infected versus uninfected children, there was a positive relationship between microbial abundance and schistosome infection intensity. Hence, it is possible that schistosome infection resulted in alterations in the gut microbiome. However as *S. haematobium* worms mostly reside in the pelvic venous plexus (although some have occasionally been detected in the intestine in Egyptian autopsies[37]), the effect of infection on the diversity of the microbiota is as suggested for intestinal helminths[33], but most likely through a more indirect or systemic route than through direct interactions[18]. Mishra et al.[18] have suggested that the immunomodulatory effects of helminths can extend to the gut microbiota through both direct intestinal interactions and systemic interactions. For example, by enhancing the mucosal barrier, tissue repair, production of antimicrobial peptides and reducing dissemination of microbiota to the spleen and liver[18], the upregulation of IL-22 during helminth infection may favour the abundance of specific microbial taxa[67].

We identified 262 AMR genes, most of which encoded for resistance to tetracycline, beta-lactam, macrolide and sulfonamide, posing risks to successful treatment of various conditions

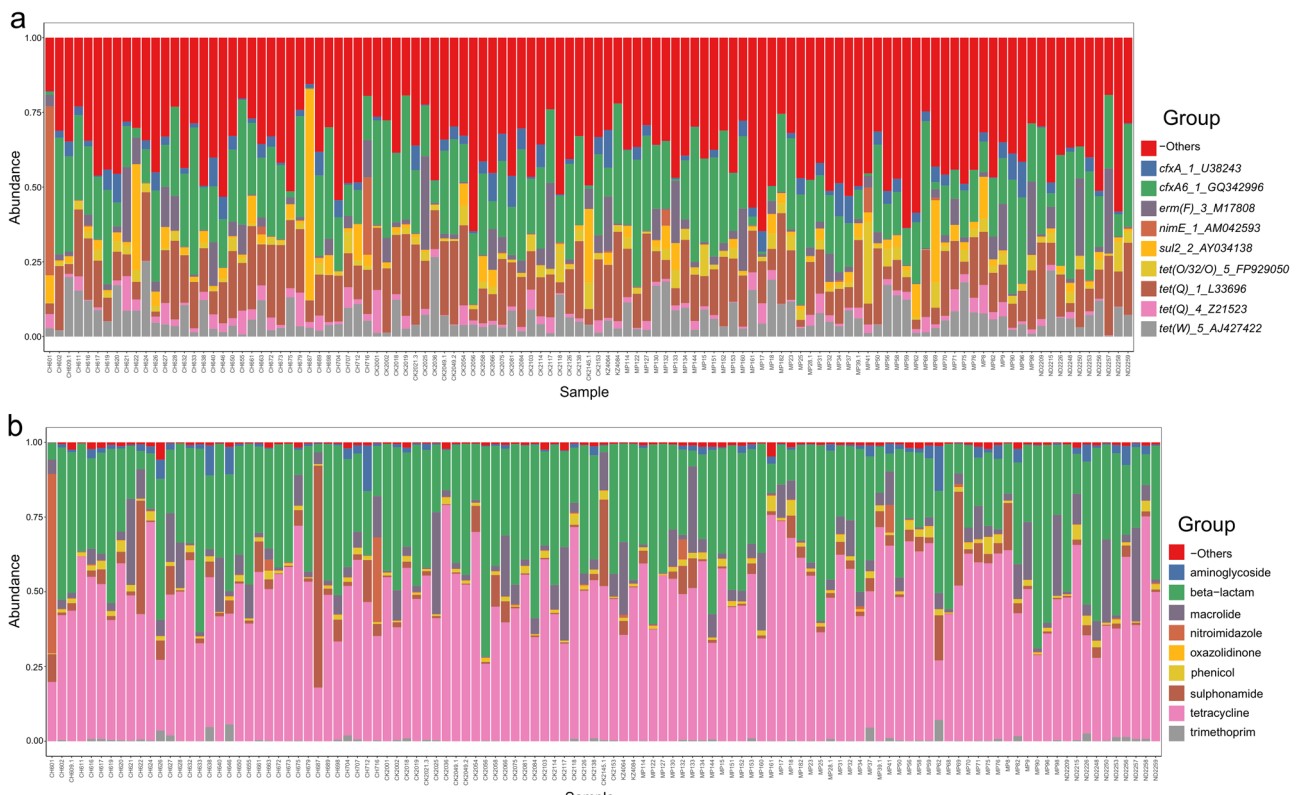

**Fig. 5 Overview of antimicrobial resistance (AMR) gene abundance and composition.** From read mapping to the ResFinder database, AMR abundance was calculated for each reference gene across all samples. Stacked bar charts show the most abundant (**a**) AMR gene and (**b**) drug class per sample, proportional to the total AMR within each sample ($n = 113$ biologically independent samples). Charts were generated using gene length-normalised, zero-corrected abundance. "-Other" represents abundance data for all other AMR genes or drug classes in the abundance data set.

including urinary, enteric and respiratory infections[68]. It is thus not surprising that the data on antimicrobial use for the current study showed predominantly amoxicillin (beta-lactam) and co-trimoxazole (sulfonamide) use in the children. In addition, ceftriaxone and benzylpenicillin (both beta-lactams) and co-trimoxazole are among the most commonly used antibiotics in Zimbabwe[69].

Increased antimicrobial use impacts the gut microbiota[70,71] and is selective for AMR in populations[72,73]. The limited association of the obtainable antimicrobial use data with both the microbiome and resistome in the current study might be surprising. However, a study by Dethlefsen et al.[70] showed that a majority of the bacterial community that was depleted post-ciprofloxacin administration was restored after 4 weeks. Our antimicrobial use data were limited to antibiotic use within the immediate 6 months prior to sampling and was less heterogeneous, thus any marked differences in the microbiota may have been missed. Our findings are consistent with those from recent studies on global sewage samples, which have shown a much stronger association between socio-economic factors related to health, sanitation and education with the resistome, compared to antimicrobial use[55,74]. This seems more likely to be the case in low- and middle-income countries, where a high contagion––the spread of resistant strains and genes––between individuals may take place, and any antimicrobial use in one individual may have general effects on the population as a whole[75]. Thus, antimicrobial use explains some, but not all variation in AMR genes in this population[72,73], and improving such socio-economic factors may improve AMR.

Our study had a few limitations. The cross-sectional study design allowed characterisation of the gut microbiome and its relationship with *S. haematobium* infection at a single time point. A longitudinal study will inform on the dynamic relationship between the two, as well as the time course or developmental-related trends in the observed profiles. A longitudinal study would also give an indication of the dynamic features of the AMR genes and how they are associated with individual-specific data. Furthermore, relating the presence of AMR genes to measurable phenotypic resistance of bacteria would give a stronger indication of the clinical implications of the AMR genes present.

In conclusion, we characterised, through shotgun metagenomic sequencing, the microbiome (to include the fungi repertoire) and resistome in a preschool population (≤5 years old) in Zimbabwe. We identified differences in the gut microbiome between schistosome infected and uninfected children, showing largely an increase in abundance of specific bacteria, and for the first time (to our knowledge), fungi genera in infected children. This association was independent of age, sex and village. Mechanistic studies are required to further explain this relationship. To the best of our knowledge, we also characterised for the first time in this African preschool population, a diversity of AMR genes to bacteria, belonging to various functional drug classes. Our microbiome and resistome data add to publicly available data from different human populations.

## Methods

**Ethical approval and consent.** The current study is part of a larger paediatric schistosomiasis study, for which ethical and institutional approval was obtained from the Medical Research Council of Zimbabwe (MRCZ/A/1964) and the

University of Edinburgh (fmutapi-0002), respectively. Permission to conduct the study was obtained from the Mashonaland Central Provincial Medical Director. The study aims and procedures were explained to all participants and their parents/guardians in their local language, Shona. Written informed consent was obtained from the participants' parents/guardians and recruitment was voluntary with participants free to withdraw from the study at any stage. For the current study, only samples from participants who consented to be part of this sub-study were used.

**Study design, population and site**. This cross-sectional study forms part of the baseline survey of a larger study on paediatric schistosomiasis conducted in the Shamva district, Northeast Zimbabwe, to compare the health benefits of early treatment of schistosome infections in preschool-aged children, 6 months to 5 years old[27]. In brief, children within this age group who were lifelong residents, and with no history of tuberculosis, malaria/fever, or recent major illness/surgery were invited to participate in the baseline survey and onward recruitment into the main cohort, if they met further criteria of being negative for *S. haematobium* by egg count and had no history of antihelminthic treatment. The study followed two groups of schistosome-negative children for new infections and then compared re-infection rates across two different regimens, following treatment of first schistosome infection.

For the current study, a subset of 116 stool samples (from 1–5 year olds) were selected from the baseline survey for microbiome analysis. To be recruited as part of this subset, participants had to consent for their stool samples to be used as part of the current study. A questionnaire was administered at the time of recruitment to gather metadata on socio-demography, growth and nutrition, and clinical history. Clinical records were checked to obtain history of antibiotic use, within the 6 months preceding acquisition of stool samples. Parents/carers were also interviewed to ascertain the health history of the children, and those who had any such history were excluded. Children were tested via parasitological diagnosis for *S. mansoni* and soil-transmitted helminths (STH) as part of the baseline survey and none who were positive were included in the subset of 116 children in the current study. The subset of children was thus selected based on (a) consent for microbiome analysis, (b) availability of socio-demographic data, (c) availability of parasitology samples (urine and stool samples), (d) availability of test results and clinical history and (e) no current episode of diarrhoea (assessed by questionnaire and visual stool examination).

Samples included in the current study were from two main sites, Mupfure and Chihuri. The sites are located in the Mashonaland Central province, with 123,650 people living in a 99% rural area of 2695 km² , according to the 2012 national census[76]. The inhabitants are primarily subsistence farmers. The area has a high prevalence of *S. haematobium* (>50%) with low prevalence of *S. mansoni* and STH (<15%)[77], making it ideal for studies on the impacts of urogenital schistosomiasis.

**Sample size**. The samples used in the current study are from the baseline survey of a larger epidemiological study comparing re-infection rates across two different treatment regimens. As this is a relatively new field in human helminthology, there are limited published studies, with none focusing on the age group in the current study, i.e. 1–5 year olds. Thus there were no published baseline data to inform sample size calculations when we conducted our study. The sample size for the current study was informed by our previous study[35] and those of others[34,39] in older children with sample sizes ranging from 34–139, from which statistically significant differences were detected in the microbiome of schistosome infected versus uninfected children.

**Sample collection, processing and DNA extraction**. Urine and stool specimens were collected for parasitological diagnosis of *S. haematobium* and intestinal helminths, respectively. In summary, about 50 ml of urine sample was collected on three successive days, and a single stool sample was collected on a single day from each participant. Urine bags (Hollister 7511 U-Bag Urine Specimen Collector, Hollister Inc., Chicago, IL, USA) and disposable diapers (for stool samples) were used for sample collection in very young children. Urine samples were examined microscopically for *S. haematobium* infection following the standard urine filtration method[78], and stool samples examined microscopically using the Kato–Katz method[79], to exclude *S. mansoni* and STH (for at least one parasite egg detected). Infection intensity for *S. haematobium* was defined as the arithmetic mean egg count/10 ml of at least two urine samples collected on three consecutive days. All children who were positive for schistosome infection were treated with a single dose of praziquantel at the standard 40 mg/kg body weight, crushed and administered with squash and sliced bread[28] by local nurses.

Aliquots of stool samples in 2 ml cryotubes were stored temporarily at 2–8 °C for a maximum of 24 h prior to processing. For each stool sample, DNA was extracted using the QIAamp DNA Stool Mini Kit (QIAGEN) according to the manufacturer's instructions. To ensure sample aliquots contained purified DNA, each sample was quantified in-house at the University of Edinburgh using the Qubit fluorometer (ThermoFisher Scientific) prior to shipment for DNA sequencing.

**Library preparation and sequencing**. DNA samples were shipped on dry ice for library preparation and sequencing at the Beijing Genomics Institute (BGI, Shenzhen, China). At BGI, DNA from the stool samples was quantified using the

Qubit fluorometer (ThermoFisher Scientific) and the NanoDrop™ spectrophotometer (ThermoFisher Scientific). As a quality control measure, the integrity and purity of DNA was assessed by a 1% agarose gel electrophoresis, run at 150 V for 40 min; DNA was sheared by ultrasonication into fragments (Covaris). Fragments were mixed with End Repair mix (BGI) and purified using the QIAquick PCR Purification Kit (Qiagen). Adapter-ligated DNA fragments were separated by electrophoresis through a 2% agarose gel to recover the target fragments, and purified using the QIAquick Gel Extraction kit (Qiagen). Library preparation to enrich the adapter-ligated DNA was done via PCR amplification, size-separated by electrophoresis, and purified using the QIAquick Gel Extraction kit (Qiagen). The final library was quantified using the Agilent 2100 bioanalyzer. The qualifying 116 DNA libraries were amplified using the cBOT system (Illumina), and sequenced on the Illumina Hiseq 4000 platform (Illumina) using paired-end 150-bp sequencing.

**Bioinformatics processing: quality control and trimming**. Raw FASTQ format sequences from each sample were quality assessed using FASTQC v0.10.0 (https://www.bioinformatics.babraham.ac.uk/projects/fastqc/). The number of read pairs generated per sample ranged from 9,263,538 to 21,350,613 [Supplementary Data 1]. Subsequently, reads were trimmed, to include removing adaptors, using BBduk2 [BBMap—Bushnell B.—https://sourceforge.net/projects/bbmap/] with an output quality Phred threshold score of ≥20 and a minimum read length of 50 bp. *K*-mer length for finding contaminants was *k* = 19. We looked for shorter *k*-mers at read pairs down to *k* = 11, and reads were trimmed at the right end [Supplementary Data 1].

**Bioinformatics processing: mapping of sequence reads**. Using a novel reference based mapping and alignment tool, *k*-mer alignment (KMA)[80], the trimmed reads were used as input to align directly against reference sequence databases. The KMA method is designed to improve mapping against redundant databases, and has been shown to outperform existing mapping methods in terms of speed, precision and sensitivity[80]. In summary, KMA, employs heuristic mapping, which involves directly mapping *k*-mers between query sequences and selected template databases, including large redundant databases. It then speeds up mapping by using *k*-mer seeding, and utilises a special version of the Needleman–Wunsch algorithm[81] to accurately align regions of mismatching *k*-mers. To ensure the best match template for the query reads, multi-mapping reads are resolved using a novel sorting scheme, ConClave. The scheme enables assembly of reads which results in a final accurate consensus sequence for the reference sequence, and also rules out bias associated with base calling across different sequencing platforms[82].

**Bioinformatics processing: microbiome sequence component**. To access the microbiome sequence component present in our samples, read pairs and singletons were aligned to a custom reference genomic database (last updated 04.04.2019). Mapped reads were counted as one copy, in cases of read pairs or singletons. Unless otherwise specified below, databases were primarily downloaded via NCBI GenBank clade specific assembly_summary.txt files (ftp://ftp.ncbi.nlm.nih.gov/genomes/genbank). The custom database consisted of the following: bacteria (closed genomes; downloaded 05.02.2019), archaea (downloaded 13.02.2019), MetaHitAssembly (PRJEB674–PRJEB1046; downloaded 01.07.2014), HumanMicrobiome (genome assemblies; downloaded 02.07.2014), bacteria_draft (downloaded 05.02.2019), plasmid (downloaded 05.02.2019), virus (https://bitbucket.org/genomicepidemiology/kvit_db; downloaded 05.02.2019; https://genome.jgi.doe.gov/portal/pages/dynamicOrganismDownload.jsf?organism=IMG_VR; downloaded 28.01.2019), fungi (downloaded 13.02.2019), protozoa (downloaded 13.02.2019), and parasites (downloaded 04.04.2019). Sequences selected for the bacteria and bacteria_draft databases from the assembly_summary.txt file were annotated with the tags version_status = "latest" and genome_rep = "Full". Additional entries, assembly_level = "Complete genome" or "Chromosome" in the bacteria database and refseq_category = "representative genome" in the bacteria_draft database were also required. The plasmid database was constructed as a subset of the bacteria and bacteria_draft sequences; keyword in the FASTA entry header line, "plasmid". The total read count for each microbial community of interest in a sample was calculated as the sum of read counts from each of the databases of interest; bacteria (bacteria, bacteria_draft, MetaHitAssembly, and HumanMicrobiome), fungi, protozoa, and parasites. Sequence mapping statistics are shown in Supplementary Data 1.

The primary (most similar) alignment obtained for mapped sequences was used to assign a putative taxonomy, based on the taxonID obtained. TaxonID's and associated taxonomy classifications were obtained from downloaded reference microbial genomes from NCBI (ftp.ncbi.nih.gov/pub/taxonomy/taxdump.tar.gz) and assignment at all taxonomic levels was done. Sequences that had no similarities detected in the nucleotide (nt) database for which we could not assign a putative taxonomic classification were deemed to be unknown sequences, and hence termed "Unknown". The classification of "unknown" is exquisitely time-sensitive, but was appropriate and correct at the time of this analysis. Details of taxonID mapping are shown in Supplementary Data 1.

**Bioinformatics processing: AMR gene component**. To identify any putative AMR genes present in the samples, the read pairs were aligned to AMR genes (3081

genes) present in the ResFinder database (https://bitbucket.org/genomicepidemiology/resfinder_db; downloaded 13.02.2019) with parameters set for a query gene to cover at least 2/5 the length of the reference gene to be selected[83]. Alignments were filtered to retain those exhibiting a selected threshold of identity of 90% (i.e. >90% nucleotide identity between the query and reference gene over at least 90% of the length of reference gene). Sequence mapping statistics are shown in Supplementary Data 1.

**Data handling and processing**. To account for probable sample-wise sequencing depth differences, as well as a size-dependent probability of observing a reference, mapping counts from the custom genomic database and from the ResFinder database were normalised to the total genome sizes for the genomic database and to the individual gene lengths for the ResFinder database (gene and genome size details in Supplementary Data 1).

The total observed mapping counts are relative, and may account for confounding effects on downstream analyses[84]. This may be due to limitations of an arbitrary total imposed by different sequencing platforms, technical variations in sequencing libraries amounts, or even random variation[85]. To obtain information about the abundances of features in our data set relative to each other, datasets were treated as compositional[85]. Data were transformed using the log-ratio approach as introduced by Aitchison[86], to make the data symmetric, linear and in a log-ratio coordinate space. However compositional methods such as this do not account for the presence of zeros associated with abundance datasets. To address this, a small pseudocount of half the smallest non-zero abundance per feature was added to each respective feature for all the normalised abundance matrices, prior to transformations[87]. Microbiota abundance data tables with counts, $x$, and $k$ number of populations (taxa members), were centred log ratio (clr) transformed, defined as[87]

$$\mathrm{clr}(x_1, \cdots, x_k) = \left( \log\left(\frac{x_1}{g_{(x)}}\right), \cdots, \left(\log\left(\frac{x_k}{g(x)}\right)\right) \right),$$

where, $g(x) = (\prod x_i)^{1/k}$ is the geometric mean of the particular composition.

AMR gene abundances were additive log-ratio (alr) transformed, taking the bacterial component of the microbiome ($x_k$) as the reference as[87]

$$\mathrm{alr}(x_1, \cdots, x_k) = \left( \log\left(\frac{x_1}{x_k}\right), \cdots, \left(\log\left(\frac{x_{k-1}}{x_k}\right)\right) \right).$$

Unless otherwise stated, clr and alr matrices were used for all downstream analyses. Raw mapping count data and their corresponding alr and clr values for the analysed samples can be found in Supplementary Data 2.

**Visualisation**. Data visualisation was performed within the R environment (www.bioconductor.org; www.r-project.org). Bar plots from normalised, zero-corrected abundance matrices were used to give an overview of the microbiota and AMR gene abundances across all samples. For cluster dendrograms, the Aitchison distance (Euclidean distance) was calculated using clr-transformed abundance data, and samples clustered based on distances (complete-linkage-clustering). To explore underlying variabilities in the microbiota and AMR genes across the data set, clr-transformed abundance data for each matrix, centred on the geometric mean, scaled by the total variance were ordinated using PCA[87], based on eigenvectors and eigenvalues[88]. The PCA involves using multivariate data reduction techniques through linear combinations of variables (principal components), each of which explains a percentage variation[89]. Box plots were used to highlight differences in microbiota abundance between two groups, and scatter plots to show the relationship between schistosome infection intensity and microbiota abundance.

**Statistics and reproducibility**. Statistical analyses were performed using various Bioconductor packages within the R environment (www.bioconductor.org; www.r-project.org). To test whether sample-related metadata predict within-group dispersion of the microbiota and the AMR genes, the Euclidean distances were calculated, using the R/Bioconductor package vegan[90]. The effect of such metadata on sample dissimilarities were determined using permutational multivariate analysis of variance (PERMANOVA; adonis2 function in the vegan package) using $P < 0.05$ as the significance threshold. An FDR (Benjamini–Hochberg FDR) correction was applied to counteract multiple testing[91].

To investigate further how specific taxa composition vary across the statistically significant metadata (from PERMOANOVA), while controlling for other variables of interest, analysis of composition of microbiomes (ANCOM) was used[92]. ANCOM was the preferred choice because it does not make any distributional assumptions of the data. The algorithm computes log-ratios of raw count data (clr), where the normalising reference value is the abundance of all remaining taxa, taken one at a time. ANCOM uses bootstrapped intervals to perform hypotheses tests while maintaining the Benjamini–Hochberg FDR set at 0.05 (ref. [91]). A taxa member was considered varying in composition across an independent variable of interest when it varied across the independent variable of interest with respect to 80% of the rest of the taxa in the data set ($W$-statistic cutoff: 0.80). By definition, the $W$ value generated (the number of times the null hypothesis is rejected for a given taxonomic group) is the ratio of a specific taxonomic group and a number of other groups (i.e. the $W$ value) that are different across two groups. The ANCOM

test for the influence of *S. haematobium* infection was controlled for age, sex and village.

As ANCOM only provides a list of taxa that vary in composition, the magnitude and direction of associations of taxa that vary in composition across independent variables was further determined. Box plots stratified by specific independent variables, using the clr-transformed abundance data of taxa previously identified as statistically significant by ANCOM were used to highlight differences in groups. To determine how these taxa varied with schistosome infection intensity, clr-transformed abundance data were regressed on the log transformed infection intensity ($\log_{10}$ [egg count + 1]).

The sample size used in the current study was based on availably of stool samples from the subset of children who gave consent and met the required selection criteria. Three samples were excluded from the overall analysis using a predefined criteria. To appropriately explain variations in the data, samples with non-missing data from at least one variable metadata from growth and nutritional indices, schistosome infection status, previous schistosome treatment and antibiotic use data (see Supplementary Data 1) were used for all downstream analysis (i.e. $n = 113$). Duplicate samples collected from two participants were used as biological/technical replicates for shotgun metagenomic sequencing.

**Reporting summary**. Further information on research design is available in the Nature Research Reporting Summary linked to this article.

## Data availability
Raw sequence data files from all 116 samples and associated metadata used in the current study are deposited in the Sequence Read Archive (SRA) of the National Centre for Biotechnology Information (NCBI) database under the BioProject accession number PRJNA521455. In Supplementary Data 1, we present sample metadata and all summary statistics generated from the analyses of sequence reads. The source data underlying statistical analyses and figures are shown in Supplementary Data 2. All other data are available on request to the corresponding author.

## Code availability
The updated R codes used for analysis of composition of microbiomes (ANCOM) are available on https://github.com/zellerlab/crc_meta/blob/master/src/ANCOM_updated.R OR from the author's webpage at https://sites.google.com/site/siddharthamandal1985/research.

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

## Acknowledgements

We thank the local nurses, village health workers and community nurses for their help with the fieldwork. Special thanks to all study participants and their parents/guardians. We also thank members of the Understanding Bilharzia project in Zimbabwe for their technical help, and all the members of the Parasite Immuno-epidemiology Group at the University of Edinburgh for their useful comments in shaping this manuscript. Our research is supported by the Thrasher Research Fund 12440, Wellcome Trust 108061/Z/15/Z, and the Oak Foundation. This research was commissioned in part by the National Institute for Health Research (NIHR) Global Health Research programme (16/136/33) using UK aid from the UK Government. The views expressed in this publication are those of the author(s) and not necessarily those of the NIHR or the Department of Health and Social Care. D.N.M.O. is supported by the Darwin Trust of Edinburgh. C.B., T.N.P., P.M., and F.M.A. are funded by the Novo Nordisk Foundation (NNF16OC0021856: Global Surveillance of Antimicrobial Resistance).

## Author contributions

D.N.M.O., T.M., F.M.A., M.E.J.W. and F.M. conceptualised and designed the study. D.N.M.O., T.M., T.C., S.A.A., J.M. and F.M. were involved in the fieldwork, sample collection and DNA extractions. D.N.M.O. and T.C., curated the field data. D.N.M.O., J.M. and F.M. organised the sequencing. T.N.P. and A.I. carried out the bioinformatics processing. D.N.M.O., P.M., T.N.P., C.B. and A.I. analysed, produced figures and interpreted the data. D.N.M.O. prepared the draft manuscript and all authors were involved in review, editing and approval of the final version of the manuscript.

## Competing interests

The authors declare no competing interests.
