## [Peer Review File · Communications Biology]

Reviewers' comments:

Reviewer #1 (Remarks to the Author):

The major aims of this paper are to understand if the gut microbiome and resistome in preschool-aged children in Zimbabwe are associated with age, location and urogenital schistosome infection. A secondary aim appears to be to add to the repository of information on the gut microbiome in an understudied population. The question of whether the gut microbiome and resistome are different in individuals with urogenital schistosome infection in this very-young population is interesting and important. The young age of the population helps deal with the issue that changes to the microbiome as a result of helminth infection may occur at this early stage, and thus not be accurately reflected by helminth infection status at a later age (when researchers more commonly collect samples from study participants to look at the microbiome and helminth infection). The fact that metagenomic shotgun sequencing was used adds value to this study as well. Clinical records of antibiotic use are another strength.

I'm not convinced that the general characterization of the microbiota of these people, and description of how this varies by location/age is as important as the schistosome part. You mention in the abstract that the gut microbiome and resistome of African children is poorly characterized, but do not give evidence showing this gap in the introduction. I'm sure there's something you could add to demonstrate this gap; for example what proportion of the total number of publicly available microbiome sequencing runs is from Africans? African children? I'm assuming it's small, but I think it's the author's responsibility to prove that the gap is large enough to make it worth showing Figure 1 and Figure 2, if they want to show it (rather than put it in the supplement). Also, given that we already know (at least in Western populations) that the microbiome varies by location and age, I would have thought that mentioning this variation mainly served to set up your decision to control for these variables in later analyses of the impact of schistosome infection. It seems to me like too much time is spent on these variations in and of themselves. Why does the reader specifically care that there are three specific taxa that are more common in Mupfure? Is that important enough to be in the abstract?

I think that the argument in the abstract that "anthropogenic factors greatly drive the resistome" seems overblown (or at least, confused me) if it is based solely on the lack of association of the resistome with individual-specific data.

You might mention early on that you did metagenomic shotgun sequencing, in case readers mistakenly think you did 16s sequencing (second line of introduction).

Formatting:

- Don't accidentally italicize and in abstract
- Many of the words in the figures are too small to read

You repeatedly say that you are using "more stringent" PERMANOVA. What do you mean by this? Stringent in comparison to clustering based on the PCA? Though PERMANOVA showed a significant effect of age whereas PCA did not, so in that sense, PCA is more stringent. I think another word besides stringent would be helpful.

I think you need more information about what you mean about establishing directionality. In line 527, you mention that the directionality assessment is a strength of this study. But, I don't think anyone just looks at ANCOM (or other) results and doesn't check which direction the change occurred in?

The statistical tools used are well-established and appropriate. The results related to the resistome, and to the impact of schistosome infection, are very interesting.

I recommend this article for publication, but feel that the authors might consider getting to the more novel sections (on schistosome infection's relationship with the microbiome and resistome) more quickly. Alternatively, they could set up the need for the earlier sections more thoroughly in the intro.

Reviewer #2 (Remarks to the Author):

This study by Osakunor and colleagues reports the results of an investigation aimed to establish associations between urogenital schistosomiasis and gut microbiome composition in pre-school children, and between the latter and the prevalence of AMR genes. The study is preliminary and descriptive in nature, but welcome given the increasing evidence of the ability of helminth parasite to modulate gut microbiome composition; the results will be of interest to the host-pathogen interaction research community.

However, some issues prevent me from recommending publication of this ms in the present form. First, given the availability of similar datasets in the literature (on *S. mansoni* for instance), a power calculation would have been appropriate. The prevalence of *S. haematobium* infection in this area was (thankfully) relatively low, so the authors were left with a limited sample size to investigate this association. Similarly, I feel the reader is left with incomplete information that inevitably casts doubts on the validity of the findings - were these children diagnosed with other helminth and non-helminth infections? The reader is referred to a previously published paper to gather this information and this should not be the case.

I have looked at said paper and no information is provided on prevalence of other helminth infections other than *S. mansoni* and non-helminth infections altogether. Unfortunately, the lack of this information and other regarding the status of individual samples (normal? diarrhoeal? etc.), the robustness of the findings becomes rather blurry.

Minor comments

Introduction:

The rationale behind the study is, in my opinion unclear. The transitions between paragraphs seem forced. It almost sounds like that the authors had access to these (undoubtedly precious) samples and decided to design a study around them, rather than the other way round. Please clarify the links between helminth-microbiome interactions, AMR and urogenital schistosomiasis in particular.

Lines 68-70: The key functions of the gut microbiome for vertebrate physiology and immune-defence have been clearly demonstrated - I would rephrase.

Materials and methods

Lines 139-141: I realise details regarding this study have been published elsewhere; however, reviewers and readers should have access to (at least) basic information on sample collection and processing, including whether faecal samples had been tested for other helminthiasis (which I gather they have), protozoan and/or non-parasitic infections, whether any sample was diarrhoeal etc.

Results

Following from the comment above, no information on whether any children harboured other parasites or non-parasitic pathogens is available.

Lines 90-1: English needs revising.

Lines 95-102. As Above.

16.01.2020

The Editor
Communications Biology

RE: RE: SUBMISSION OF REVISED MANUSCRIPT (COMMSBIO-19-1562A)

We thank the reviewers for their very constructive and helpful comments in clarifying and improving our manuscript with revised title **“The gut microbiome but not the resistome is associated with urogenital schistosomiasis in Zimbabwean preschool-aged children”**. The comments and suggestions have helped shape the manuscript to improve clarity, precision and quality. We have detailed below how we have revised the manuscript in response to each of the comments and suggestions from the reviewers.

RESPONSE TO REVIEWER COMMENTS

From Editor

Data Availability

Complete information about our policies on the availability of data, materials and methods can be found on our policies page. All Communications Biology manuscripts must include a section titled "Data Availability" at the end of the Methods section or main text (if no Methods). See here for more information on this policy and a list of examples.

Response: This has been moved to the end of the methods section, subheading “data availability”

REVIEWER #1 (REMARKS TO THE AUTHOR):

1. The major aims of this paper are to understand if the gut microbiome and resistome in preschool-aged children in Zimbabwe are associated with age, location and urogenital schistosome infection. A secondary aim appears to be to add to the repository of information on the gut microbiome in an understudied population. The question of whether the gut microbiome and resistome are different in individuals with urogenital schistosome infection in this very-young population is interesting and important. The young age of the population helps deal with the issue that changes to the microbiome as a result of helminth infection may occur at this early stage, and thus not be accurately reflected by helminth infection status at a later age (when researchers more commonly collect samples from study participants to look at the microbiome and helminth infection). The fact that metagenomic shotgun sequencing was used adds value to this study as well. Clinical records of antibiotic use are another strength.

- ***Response: We appreciate the reviewer’s positive comments on this.***

2. I’m not convinced that the general characterization of the microbiota of these people, and description of how this varies by location/age is as important as the schistosome part. You mention in the abstract that the gut microbiome and resistome of African children is poorly characterized, but do not give evidence showing this gap in the introduction. I’m sure there’s something you could add to demonstrate this gap; for example what proportion of the total number of publicly available microbiome sequencing runs is from Africans? African children? I’m assuming it’s small, but I think it’s the author’s responsibility to prove that the gap is large enough to make it worth showing Figure 1 and Figure 2, if they want to show it (rather than put it in the supplement). Also, given that we

already know (at least in Western populations) that the microbiome varies by location and age, I would have thought that mentioning this variation mainly served to set up your decision to control for these variables in later analyses of the impact of schistosome infection. It seems to me like too much time is spent on these variations in and of themselves. Why does the reader specifically care that there are three specific taxa that are more common in Mupfure? Is that important enough to be in the abstract?

- **Response: We accept the reviewer's suggestion. We have improved the introduction to highlight the gaps on the characterisation of the gut microbiome and resistome of African populations, especially young children. Previous figure 1 has been maintained and separated into bacteria and fungi (now as figure 1 and 2 respectively) to make the figure more legible. Previous figure 2 (ANCOM analyses by age and village), as suggested has now been moved to the supplementary as it indeed set up a justification to control for these variables in determining the association with schistosome infection.**

3. I think that the argument in the abstract that "anthropogenic factors greatly drive the resistome" seems overblown (or at least, confused me) if it is based solely on the lack of association of the resistome with individual-specific data.

- **Response: We accept the reviewer's suggestion. Based on the manuscript preparation checklist, we have had to reduce the abstract considerably to the most relevant information; this has been removed from the abstract and also in the main conclusions, and only maintained in the discussion. As suggested in the discussion, this is in line with recent studies on global sewage samples, which have shown a much stronger association between socio-economic factors related to health, sanitation, and education with the resistome, compared to AMU (Collignon, Beggs et al. 2018, Hendriksen, Munk et al. 2019). And that AMU explains some, but not all variation in AMR genes in this population (Van De Sande-Bruinsma, Grundmann et al. 2008, Davies and Davies 2010), hence confirmed the fact that we did not find any association between AMR and AMU and such factors may be driving variations in communities. This section has thus been rephrased to reflect that these are suggestions as found by other studies.**

4. You might mention early on that you did metagenomic shotgun sequencing, in case readers mistakenly think you did 16s sequencing (second line of introduction).

- **Response: We have done this as suggested.**

Formatting:

5. Don't accidentally italicize and in abstract

- **Response: We appreciate the reviewer's feedback on this. The manuscript has been proof read carefully and all accidental italics have been corrected.**

6. Many of the words in the figures are too small to read

- **Response: We appreciate the reviewer's recommendation on this. All figures have been formatted and checked for text legibility.**

7. You repeatedly say that you are using "more stringent" PERMANOVA. What do you mean by this? Stringent in comparison to clustering based on the PCA? Though PERMANOVA showed a significant effect of age whereas PCA did not, so in that sense, PCA is more stringent. I think another word besides stringent would be helpful.

- ***Response: We appreciate the reviewer's suggestion on this. We used the PCA as an ordination technique to reveal hidden patterns in the data (assuming linear relationships between taxa and the underlying ecological gradients (and this is often not the case) hence our observation. We have removed the word stringent from these and any of such sentences and kept them straight to the point.***

8. I think you need more information about what you mean about establishing directionality. In line 527, you mention that the directionality assessment is a strength of this study. But, I don't think anyone just looks at ANCOM (or other) results and doesn't check which direction the change occurred in?

- ***Response: We agree with the reviewer's suggestion on this. We have removed this part from this and other similar sentences..***

9. The statistical tools used are well-established and appropriate. The results related to the resistome, and to the impact of schistosome infection, are very interesting.

- ***Response: We appreciate the reviewer's positive comments on this.***

I recommend this article for publication, but feel that the authors might consider getting to the more novel sections (on schistosome infection's relationship with the microbiome and resistome) more quickly. Alternatively, they could set up the need for the earlier sections more thoroughly in the intro.

- ***Response: We accept the reviewer's suggestion. We have set up the need for the earlier sections on characterisation in the intro more thoroughly. We have also revised the results section to focus more on the novel sections on schistosome infection's relationship with the microbiome and resistome (as raised in the earlier comment above). The section on characterisation has been limited to the overview of the microbiome and resistome, with the separate analysis by age and village moved to the supplementary.***

REVIEWER #2 (REMARKS TO THE AUTHOR)

This study by Osakunor and colleagues reports the results of an investigation aimed to establish associations between urogenital schistosomiasis and gut microbiome composition in pre-school children, and between the latter and the prevalence of AMR genes. The study is preliminary and descriptive in nature, but welcome given the increasing evidence of the ability of helminth parasite to modulate gut microbiome composition; the results will be of interest to the host-pathogen interaction research community.

- ***Response: We appreciate the reviewer's positive comments on this.***

1. However, some issues prevent me from recommending publication of this ms in the present form. First, given the availability of similar datasets in the literature (on *S. mansoni* for instance), a power calculation would have been appropriate. The prevalence of *S. haematobium* infection in this area was (thankfully) relatively low, so the authors were left with a limited sample size to investigate this association.

- ***Response: We accept the reviewer's concern on omitting such information. We have included a subsection under the methodology, "Sample size" to clarify this in detail. The samples in the current study are from the baseline survey of a larger epidemiological study comparing re-infection rates across two different treatment regimens. As this is a relatively new field in human helminthology, there are limited published studies, with***

none focusing solely on the age group we focused on in the current study, i.e. 1-5 year olds. Thus there were no published baseline data to inform sample size calculations when conducted our study. The sample size for the current study was informed by our previous study (Kay, Millard et al. 2015) and those of others (Schneeberger, Coulibaly et al. 2018, Ajibola, Rowan et al. 2019) in older children with sample sizes ranging from 34–139, from which significant differences were detected in the microbiome of schistosome infected versus uninfected children.

2. Similarly, I feel the reader is left with incomplete information that inevitably cast doubts on the validity of the findings - were these children diagnosed with other helminth and non-helminth infections? The reader is referred to a previously published paper to gather this information and this should not be the case.

I have looked at said paper and no information is provided on prevalence of other helminth infections other than *S. mansoni* and non-helminth infections altogether. Unfortunately, the lack of this information and other regarding the status of individual samples (normal? diarrhoeal? etc.), the robustness of the findings becomes rather blurry.

- ***We accept the reviewer’s suggestion. We have included a paragraph detailing the criteria for data collection and inclusion in the current study under “Study design, population and site”. This includes; a) consent for samples to be used for microbiome analysis, b) the availability of complete socio-demographic data, c) all parasitology samples (urine and stool samples), d) availability of test results and clinical history (inclusion based on being negative for *S. mansoni* and soil transmitted helminths (STH), no recent history of illness), and e) no current episode of diarrhoea (also ascertained through questionnaires and visual or macro assessment of stool samples). We have also included a brief description of the parent study from of which this is a subset. As part of the inclusion criteria of the larger study (stated in this section also) was to have no clinical symptoms of tuberculosis, malaria/fever, or recent major illness/operation. Children were also tested via parasitological diagnosis for *S. mansoni* and soil-transmitted helminths (STH) as part of the baseline survey and no *S. mansoni* or STH –positive children were included in the current study. We have also included details of questionnaire administration, and the use of clinical records in addition to interviewing parents and care givers to ascertain the health history of the children and to exclude any recent illness/surgery, and malaria/fever.***

We have also included a paragraph on sample collection and parasitological diagnosis under the section “Sample collection, processing and DNA extraction”, detailing parasitological diagnosis of *S. haematobium* from urine, the exclusion of intestinal schistosomiasis and Soil transmitted helminths (STH) from stool examination, and treatment of schistosome positive children.

Data to support this has been included in this section highlighting the high prevalence of *S. haematobium* but low prevalence of *S. mansoni* and STH, making it ideal for our study on urogenital schistosomiasis.

Minor comments

3. Introduction:

The rationale behind the study is, in my opinion unclear. The transitions between paragraphs seem forced. It almost sounds like that the authors had access to these (undoubtedly precious) samples

and decided to design a study around them, rather than the other way round. Please clarify the links between helminth-microbiome interactions, AMR and urogenital schistosomiasis in particular.

- **Response: We accept the reviewer's suggestion. We have revised the entire introduction to reflect thoroughly the rationale behind the study with better transitions throughout sections. We have provided evidence clarifying the need to characterise the microbiome of this population, links between helminth-microbiome interactions, and finally, in addition to the schistosome-microbiota interactions, other interactions that occur within the gut ecosystem and how a change in the microbiota diversity may impact indirectly on that of AMR genes (relevant links and examples of such interactions have been included in the introduction- penultimate paragraph).**

4. Lines 68-70: The key functions of the gut microbiome for vertebrate physiology and immune-defence have been clearly demonstrated - I would rephrase.

- **Response: We accept the reviewer's suggestion. This has been rephrased to reflect this.**

Materials and methods

5. Lines 139-141: I realise details regarding this study have been published elsewhere; however, reviewers and readers should have access to (at least) basic information on sample collection and processing, including whether faecal samples had been tested for other helminthiases (which I gather they have), protozoan and/or non-parasitic infections, whether any sample was diarrhoeal etc.

- **Response: We accept the reviewer's suggestion. As above in comment #2, we have added sections and paragraphs in the "Methods" that address and clarify these.**

Results

6. Following from the comment above, no information on whether any children harboured other parasites or non-parasitic pathogens is available.

- **Response: We accept the reviewer's suggestion. As clarified, above, this was part of the exclusion criteria for recruitment into the main cohort of the parent study, hence none of such children were included in the current study. This has been clearly outlined in the methods section to reflect these details.**

7. Lines 90-1: English needs revising.

- **Response: We accept the reviewer's suggestion. This sentence has been revised to improve clarity**

8. Lines 95-102. As Above.

Response: We accept the reviewer's suggestion. This sentence has been revised to improve clarity

REVIEWERS' COMMENTS:

Reviewer #1 (Remarks to the Author):

The authors have very comprehensively addressed all my concerns. Thank you for your genuine and thoughtful responses and changes. My only remaining suggestion is that the figures could still have slightly larger font. But if the figures pass the journal's requirements for font size, then I guess they are fine! Congrats on an excellent study and paper.

Reviewer #2 (Remarks to the Author):

The authors have satisfactorily addressed my concerns. The only additional element I would point out regarding the selection criteria for the study population recent use of antibiotics (other than amoxicillin) - was this also part of the 'screening' procedures? Otherwise, I think that the current manuscript is suitable for publication in Communications Biology.

Response to reviewer comments

Reviewer #1 (Remarks to the Author):

The authors have very comprehensively addressed all my concerns. Thank you for your genuine and thoughtful responses and changes. My only remaining suggestion is that the figures could still have slightly larger font. But if the figures pass the journal's requirements for font size, then I guess they are fine! Congrats on an excellent study and paper.

Response: We have updated the font sizes

Reviewer #2 (Remarks to the Author):

The authors have satisfactorily addressed my concerns. The only additional element I would point out regarding the selection criteria for the study population recent use of antibiotics (other than amoxicillin) - was this also part of the 'screening' procedures? Otherwise, I think that the current manuscript is suitable for publication in Communications Biology.

Response: No this wasn't a part of the criteria, and we did not state this as part of the selection criteria. However, data on general use of antibiotics within the stated period (past 6 months) was collected and this was used as part of the data analyses